# RAFFT: Efficient prediction of RNA folding pathways using the fast Fourier transform

**Vaitea Opuu◉\*, Nono S. C. Merleau◉, Vincent Messow◉, Matteo Smerlak◉**

Max Planck Institute for Mathematics in the Sciences, Leipzig, Germany

\* vopuu@mis.mpg.de

**Data Availability Statement:** RAFFT and the benchmark data used in thisvmanuscript are available at https://github.com/strevol-mpi-mis/RAFFT. We also provide the scripts used for the figures and kinetic analyses.

## Abstract

We propose a novel heuristic to predict RNA secondary structure formation pathways that has two components: (i) a folding algorithm and (ii) a kinetic ansatz. This heuristic is inspired by the kinetic partitioning mechanism, by which molecules follow alternative folding pathways to their native structure, some much faster than others. Similarly, our algorithm RAFFT starts by generating an ensemble of concurrent folding pathways ending in multiple meta-stable structures, which is in contrast with traditional thermodynamic approaches that find single structures with minimal free energies. When we constrained the algorithm to predict only 50 structures per sequence, near-native structures were found for RNA molecules of length $\leq$ 200 nucleotides. Our heuristic has been tested on the coronavirus frameshifting stimulation element (CFSE): an ensemble of 68 distinct structures allowed us to produce complete folding kinetic trajectories, whereas known methods require evaluating millions of sub-optimal structures to achieve this result. Thanks to the fast Fourier transform on which RAFFT (RNA folding Algorithm wih Fast Fourier Transform) is based, these computations are efficient, with complexity $\mathcal{O}(L^2 \log L)$.

## Author summary

The understanding of RNA's behaviour at the molecular level is crucial for novel applications such as RNA-based vaccines or gene editing technologies. As proteins, RNA molecules fold into complex molecular structures dictated by their sequence of nucleotides. Identifying relevant molecular structures of the folding processes is essential but computationally challenging. Whereas classical approaches predict a single molecular structure, we propose a method that predicts folding trajectories by leveraging the fast Fourier transform algorithm to identify structural fragments quickly. We showed that the folding trajectories predicted reflect complementary information to classical methods while allowing us to identify biologically relevant structures.

This is a *PLOS Computational Biology* Methods paper.

**Funding:** MS is funded by Sofja Kovalevskaja Award endowed by the German Federal Ministry of Education and Research, and by the Human Science Frontier Program Organization through a Young Investigator Award grant RGY0077/2019. The funders had no role in study design, data collection and analysis, decision to publish, or preparation of the manuscript.

## Introduction

The function of noncoding RNAs is largely determined by their three-dimensional structure [1]. For instance, the catalytic function of ribozymes can often be analyzed in terms of basic structural motifs, such as hammerhead or hairpin structures [2]. Other RNAs, like riboswitches, involve changes between alternative structures [3]. Understanding the relation sequence and structure is therefore a central challenge in molecular biology. Because measuring the structure of RNAs through X-ray crystallography or NMR is difficult and expensive, computational approaches have played a central role in the analysis of natural RNAs [4, 5], and also in the design of synthetic RNAs [6].

Three levels of structures are used to describe RNA molecules: (1) the primary structure, that is, the nucleotide sequence itself; (2) the secondary structure formed by Watson-Crick (or wobble) base pairings; (3) the tertiary structure represents the molecule shape in three-dimensional space. Unlike proteins, RNA structures are usually formed hierarchically; the secondary structure is formed first, followed by the tertiary structure [7]. This separation of time scales justifies focusing on the prediction of secondary structures; evidence from molecular mechanical stretching experiments [8] suggests that the resulting tertiary structures (as well as the kinetic bottlenecks towards their formation) are indeed largely determined by the RNA's secondary structure.

Although base pairs can be formed with various configurations [9], we only consider here the canonical interactions: G-C, A-U, and G-U. Moreover, while various subtleties are involved in the definition of the secondary structure, we use here the formal definition called pseudoknot-free [10]. In the rest of this work, 'structure' refers specifically to this notion of RNA secondary structure.

The thermodynamic stability $\Delta G_s$ of a structure $s$ is the free energy difference with respect to the completely unfolded state. To predict biologically relevant structures, most computational methods search for structures that minimize this free energy. To this aim, structures are decomposed into components called loops, such that using the additivity principle [11], the free energy of a structure can be approximated by the sum of its constituent loops free energies. Many models allow to compute the free energies of those constituent loops, but the dominant one is the nearest-neighbor loop energy model [12]. This model associates tabulated free energy values to loop types and nucleotide compositions; the Turner2004 [13] is one of the most widely used set of parameters. This structure decomposition allows an efficient dynamic programming algorithm that can determine the minimum free energy (MFE) structure of a sequence in the entire structure space [14].

The MFE structure is commonly used in free-energy based predictions; however, it represents one structural estimate among many others, including the maximum expected accuracy (MEA).

Several existing tools implement the Zuker dynamic programming algorithm [14], e.g. `RNAfold` [15], `Mfold` [16], or `RNAstructure` [17]. While these methods were found to predict RNA structures accurately, as shown in recent benchmarks [18, 19], the additivity principle is expected to break down when structures are too large. Moreover, thermodynamic models tend to ignore pseudoknot loops, which can sometimes limit their biological relevance.

Recently, machine learning (ML) approaches were investigated and seemed to overcome some of these shortcomings. ML-based structure prediction tools provide substantial improvements [18, 20]. However, in addition to some over-fitting concerns [21], these approaches cannot give dynamical information, as few data are available on structural dynamics. In addition, ML methods do not follow from first principles: structural training data are mostly obtained through phylogenetic analyses. Consequently, the predictions from those methods may be biased, e.g. due to *in vivo* third-party elements.

From the dynamical standpoint, the RNA molecule navigates its structure space by following a free energy landscape. Three rate models describing elementary steps in the structure space are currently used to study RNAs folding dynamics: (1) the base stack model uses base stacks formations and breaking as elementary moves [22]; (2) the base pair model as implemented in `kinfold` [23] gives the finest resolution with base pair steps, but at the cost of computation time; (3) the stem model [24] provides a coarse-grained description of the dynamics, where free energy changes due to stem formation guide the folding process. The latter makes a notable assumption: transition states (or saddle points) involved in the formation of a stem are not considered [25]. An alternative approach, implemented in `kinwalker` [26], used the observation that folded intermediates are generally locally optimal conformations.

In folding experiments, Pan and coworkers observed two kinds of pathways in the free energy landscape of a natural ribozyme [27]. Firstly, the experiments revealed fast-folding pathways, in which a sub-population of RNAs folded rapidly into the native state. The second population, however, quickly reached metastable misfolded states, then slowly folded into the native structure. In some cases, these metastable states are functional. These phenomena are direct consequences of the rugged nature of the RNA folding landscape [28]. The experiments performed by Russell and coworkers also revealed the presence of multiple deep channels separated by high energy barriers on the folding landscape, leading to fast and slow folding pathways [29]. The formal description of the above mechanism, called kinetic partitioning mechanism, was first introduced by Guo and Thirumalai in the context of protein folding [30]. In the free energy landscape, these metastable conformations form competing attraction basins in which RNA molecules are temporarily trapped. However, *in vivo*, folding into the native states can be promoted by molecular chaperones [31], which means that the active structure depends on factors other than the sequence. This may rise some discrepancy when comparing thermodynamic modelling to real data.

Here, we propose a novel approach to RNA structure prediction and dynamics inspired by the kinetic partitioning mechanism. Our method has two components: (1) a folding algorithm that models the fast-folding pathways and (2) a kinetic ansatz that displays how the conformations are populated over time (Fig 1).

The folding algorithm constructs multiple parallel folding pathways by sequentially forming stems. This procedure yields an ensemble of structures modelling the complete folding process, from the unfolded state to multiple folded states. The FFT algorithm on which `RAFFT` is based has already been used in the analysis of sequences [32]; for example, it powers `MAFFT`, a well-known multiple-sequence-alignment tool [33].

The quality of the predicted ensembles of structures has been assessed on a the well-curated dataset `ArchiveII` [34]. The results were compared to two structure estimates: the MFE

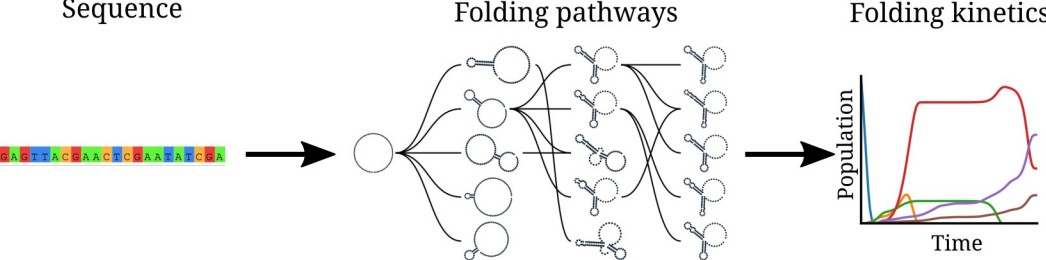

**Fig 1. RAFFT framework.** (1) The sequence of nucleotides of the RNA molecule is first encoded numerically to facilitate its treatment. (2) The numerically encoded sequence is analyzed using the FFT to detect stems quickly, such that stems are added iteratively to form parallel folding paths. (3) The folding pathways form a graph that connects potential intermediate secondary structures. Starting with the unfolded structure, our kinetic ansatz predicts a complete folding trajectory (*i.e.* when the RNA molecule adopts a specific structure).

structure computed with `RNAfold`, and the ML structure computed with `MXfold2` [18] since methods of each approaches displayed similar performances.

Using `RAFFT`, we investigated the folding kinetic of the Coronavirus frameshifting stimulation element (CFSE) [35]. `RAFFT`'s procedure displayed results qualitatively similar to the state-of-the-art barrier kinetics [23]. However, our procedure requires drastically fewer structures and models the complete folding process from the unfolded state. Our kinetic modelling revealed that the native structure of the CFSE is a kinetic trap while the MFE structure only appears some time after.

## Material and methods

### Folding algorithm

We start from a sequence of nucleotides $S = (S_1 \ldots S_L)$ of length $L$, and its associated unfolded structure. We first create a numerical representation of $S$ where each nucleotide is replaced by a unit vector of 4 components:

$$
A \to \begin{pmatrix} 1 \\ 0 \\ 0 \\ 0 \end{pmatrix}, C \to \begin{pmatrix} 0 \\ 1 \\ 0 \\ 0 \end{pmatrix}, G \to \begin{pmatrix} 0 \\ 0 \\ 1 \\ 0 \end{pmatrix}, U \to \begin{pmatrix} 0 \\ 0 \\ 0 \\ 1 \end{pmatrix}. \tag{1}
$$

This encoding gives us a $(4 \times L)$-matrix we call $X$, where each row corresponds to a nucleotide as shown below:

$$
X = \begin{pmatrix} X^A \\ X^C \\ X^G \\ X^U \end{pmatrix} = \begin{pmatrix} X^A(1) & X^A(2) & \ldots & X^A(L) \\ X^C(1) & X^C(2) & \ldots & X^C(L) \\ X^G(1) & X^G(2) & \ldots & X^G(L) \\ X^U(1) & X^U(2) & \ldots & X^U(L) \end{pmatrix} \tag{2}
$$

For example, $X^A(i) = 1$ if $S_i = A$. Next, we create a second copy $\bar{S} = (\bar{S}_L \ldots \bar{S}_1)$ for which we reversed the sequence order. Then, each nucleotide of $\bar{S}$ is replaced by one of the following vectors:

$$
\bar{A} \to \begin{pmatrix} 0 \\ 0 \\ 0 \\ w_{AU} \end{pmatrix}, \bar{C} \to \begin{pmatrix} 0 \\ 0 \\ w_{GC} \\ 0 \end{pmatrix}, \bar{G} \to \begin{pmatrix} 0 \\ w_{GC} \\ 0 \\ w_{GU} \end{pmatrix}, \bar{U} \to \begin{pmatrix} w_{AU} \\ 0 \\ w_{GU} \\ 0 \end{pmatrix}. \tag{3}
$$

$\bar{A}$ (respectively $\bar{C}, \bar{G}, \bar{U}$) is the complementary of $A$ (respectively $C, G, U$). $w_{AU}, w_{GC}, w_{GU}$ represent the weights associated with each canonical base pair; these parameters are chosen empirically. We call this complementary copy $\bar{X}$, the mirror of $X$.

To search for stems, we use the complementary relation between $X$ and $\bar{X}$ with the correlation function $\text{cor}(k)$. This correlation is defined as the sum of individual $X$ and $\bar{X}$ row

correlations:

$$\text{cor}(k) = \sum_{\alpha \in \{A,C,G,U\}} c_{X^\alpha,\bar{X}^\alpha}(k), \tag{4}$$

where a row correlation between $X$ and $\bar{X}$ is given by:

$$c_{X^\alpha,\bar{X}^\alpha}(k) = \sum_{\substack{1 \le i \le L \\ 1 \le i+k \le L}} \frac{X^\alpha(i)\bar{X}^\alpha(i+k)}{\min(k, 2L-k)}. \tag{5}$$

For each $\alpha \in \{A, C, G, U\}$, $X^\alpha(i) \times \bar{X}^\alpha(i+k)$ is non-zero if sites $i$ and $i+k$ can form a base pair, and has the chosen weight; therefore, the set of weights can be seen as a simplified energy function scoring stems. If all the weights are set to 1, $\text{cor}(k)$ gives the likelihood of base pairs for a positional lag $k$. However, the weights can be tuned in order to facilitate the detection of stems containing some types of stems, *i.e.* as G-C interactions are known to be stronger, a larger weight can be used to favor the detection of G-C rich stems. Although the correlation naively requires $O(L^2)$ operations, it can take advantage of the FFT which reduces its complexity to $\mathcal{O}(L \log L)$.

Large $\text{cor}(k)$ values between the two copies indicate positional lags $k$ where the frequency of base pairs is high; however, this does not allow to determine the exact stem positions. Hence, we use a sliding window strategy to search for the largest stem within the positional lag (since the copies are symmetrical, we only need to slide over one-half of the positional lag). Once the largest stem is identified, we compute the free energy, using the ViennaRNA package API [36], change associated with the formation of that stem. We perform this search for the $n$ highest correlation values, which gives us $n$ potential stems. Then, we define the stem with the lowest free energy as the current structure.

We are now left with two independent parts, the interior and the exterior of the newly formed stem. If the exterior part is composed of two fragments, they are concatenated into one. Then, we apply recursively the same procedure on the two parts independently in a *breadth-first* fashion to form new consecutive base pairs. The procedure stops when no base pair formation can improve the energy. When multiple stems can be formed in these independent fragments, we combine all of them and pick the composition with the best overall stability. If too many compositions can be formed, we restrict this to the $10^3$ best in terms of energy. Fig 2 shows an example of a single step to illustrate the procedure.

The complexity of this algorithm depends on the number and size of the stems formed. The main operations performed for each stem formed are: (1) the evaluation of the correlation function $\text{cor}(k)$, (2) the sliding-window search for stems, and (3) the energy evaluation. We based our approximate complexity on the correlation evaluation since it is the most computationally demanding step; the other operations only contribute a multiplicative constant at most. The best case is the trivial structure composed of one large stem where the algorithm stops after evaluating the correlation on the complete sequence. At the other extreme, the worst case is one where at most $L/2$ stems of size 1 (exactly one base pair per stem) can be formed. The approximate complexity therefore depends on $\sum_{i=0}^{L/2}(L-2i)\,\log(L-2i) = \mathcal{O}(L^2 \log L)$.

The algorithm described so far tends to be stuck in the first local minima found along the folding trajectory. To alleviate this, we implemented a stacking procedure where the $N$ best trajectories are saved in stacks and evolved in parallel. As shown in Fig 3, the algorithm starts with the unfolded structure; then, the $N$ most stable stems are saved iteratively in stacks, leading to the construction of a graph we call *fast-folding graph*. The empirical time complexity of

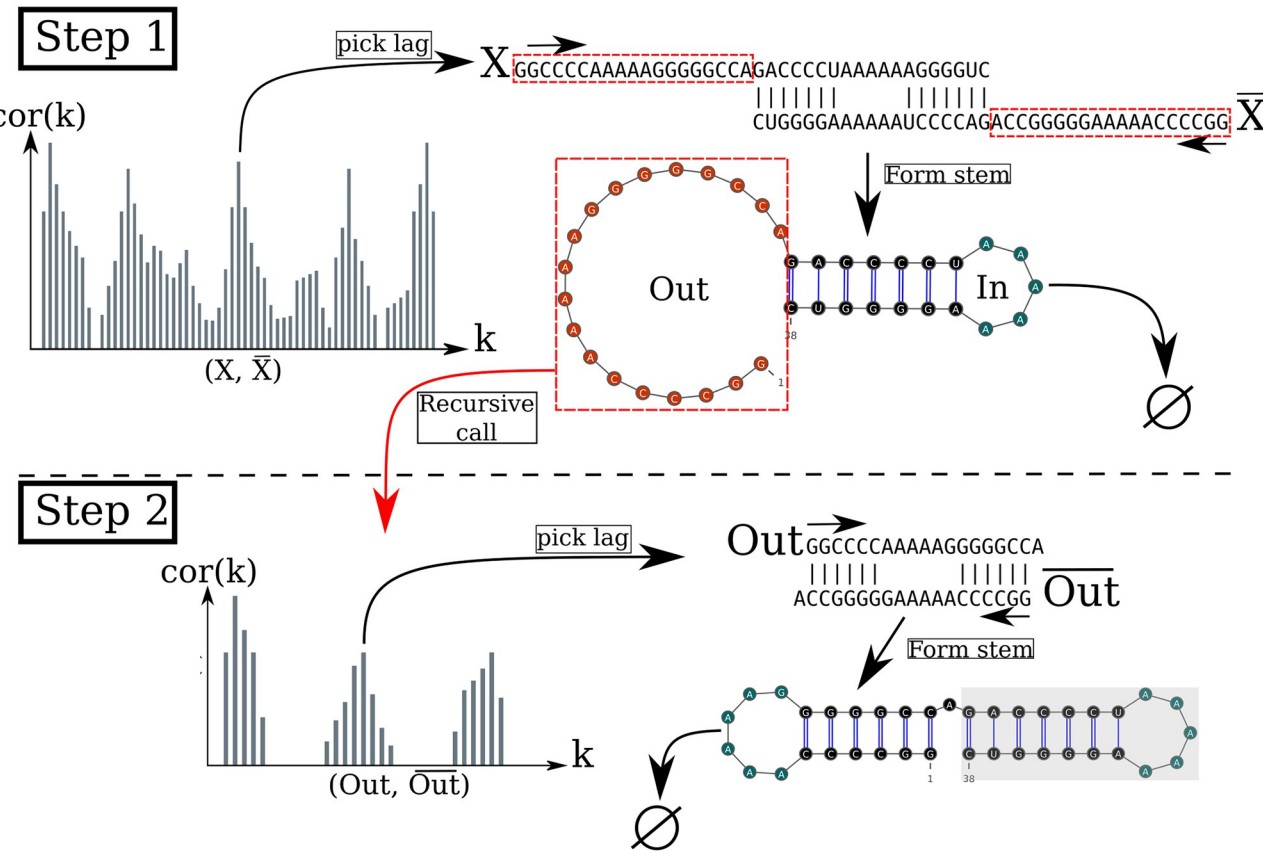

**Fig 2. Algorithm execution for one example sequence which requires two steps.** (Step 1) From the correlation $cor(k)$, we select one peak which corresponds to a position lag $k$. Then, we search for the largest stem and form it. Two fragments, "In" (the interior part of the stem) and "Out" (the exterior part of the stem), are left, but only the "Out" may contain a new stem to add. (Step 2) The procedure is called recursively on the "Out" sequence fragment only. The correlation $cor(k)$ between the "Out" fragment and its mirror is then computed and analyzing the $k$ positional lags allows to form a new stem. Finally, no more stem can be formed on the fragment left (colored in blue), so the procedure stops.

the naive algorithm and the stacked version only changes by a scaling pre-factor (Fig 4). The naive naive algorithm is very fast for sequence up to $10^4$; however, LinearFold [19] is the fastest for the largest sequences.

## Kinetic ansatz

Our folding kinetic ansatz uses the fast-folding graph to model the slow processes by which RNA molecules slowly escape from metastable structures. As described in Fig 3, transitions follows the formation or destruction of stems. The fast-folding graph follows the idea that parallel pathways quickly reach their endpoints; however, when the endpoints are non-native states, this ansatz allows slowly folding back into the native state [27].

As usually done, the kinetics is modelled as a continuous-time Markov chain [38], where populations of structures evolve according to transition rates. In this context, an Arrhenius formulation is commonly used to derive transition rates $r(x \rightarrow y) \propto \exp(-\beta E^{\ddagger})$, where $E^{\ddagger}$ is the activation energy separating $x$ from $y$. In contrast, our kinetic ansatz uses transition rates $r(x \rightarrow y)$ based on the Metropolis scheme already used in [39], and defined as

$$r(x \rightarrow y) = k_0 \times \min(1, \exp(-\beta \Delta\Delta G(x \rightarrow y))), \tag{6}$$

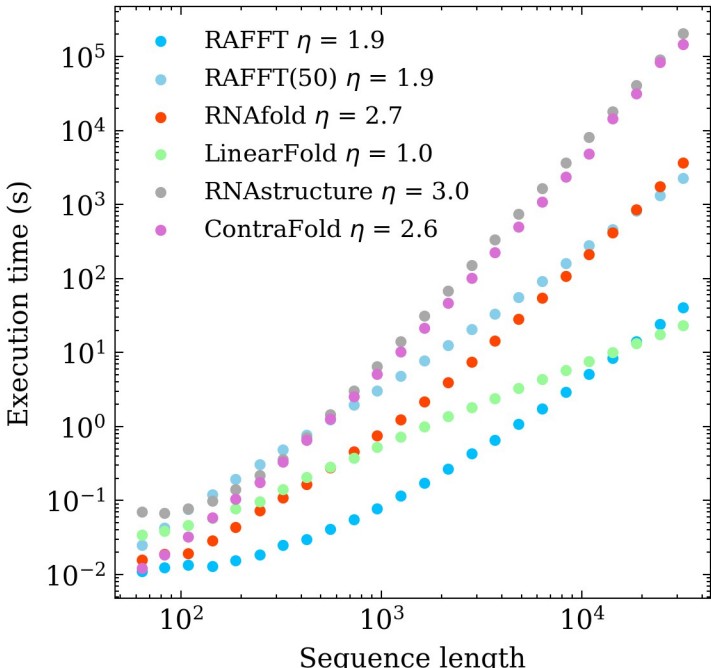

**Fig 3. Fast folding graph constructed using `RAFFT`.** In this example, the sequence is folded in two steps: starting from the unfolded structure, the $N = 5$ most stable stems found are stored in stack 1. From stack 1, multiple stems can be formed but only the $N = 5$ most stable are stored in stack 2. All secondary structure visualizations were obtained using `VARNA` [37].

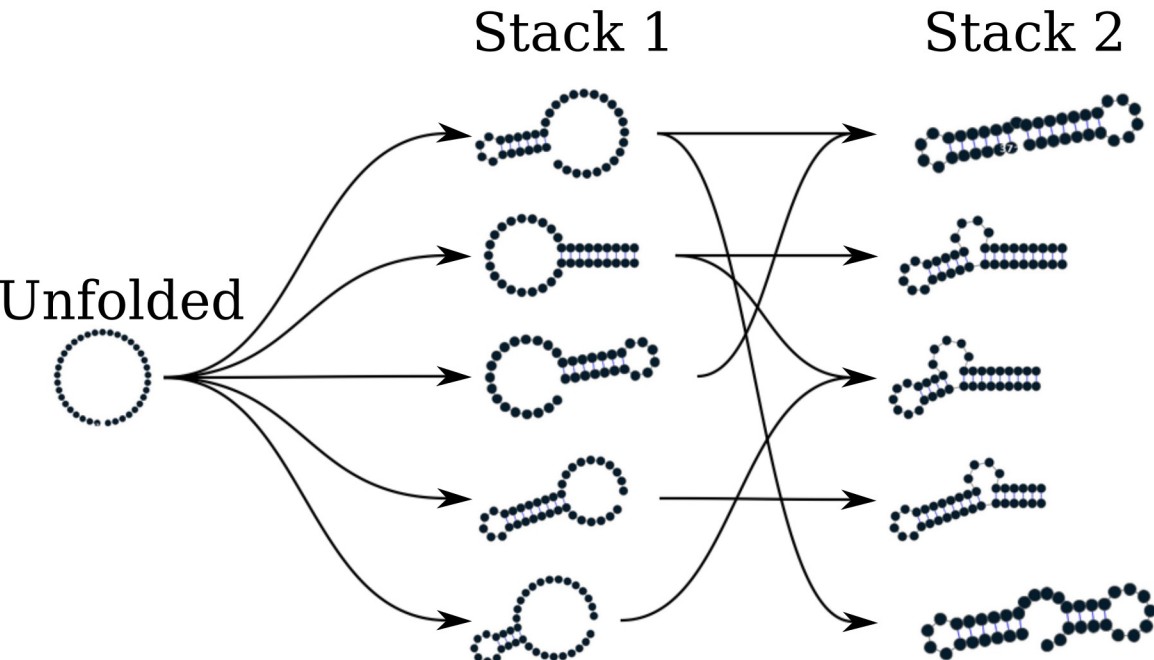

**Fig 4. Execution time comparisons.** For samples of 30 sequences per length, we averaged the execution times of five folding tools. The empirical time complexity $O(L^\eta)$ where $\eta$ is obtained by non-linear regression. RAFFT denotes the naive algorithm whereas RAFFT(50) denotes the algorithm where 50 structures can be saved per stack.

where $\Delta\Delta G(x \rightarrow y)$ is the stability change between structure $x$ and $y$. Here $k_0$ is a conversion constant that we set to 1 for the sake of simplicity. These transitions are only allowed if $y$ is connected to $x$ in the graph (i.e. $y$ is in the neighborhood of $x$, $y \in \mathcal{X}$). Here, we initialize the population $p_x(0)$ with only unfolded structures; therefore, the trajectory represents a complete folding process. The frequency of a structure $x$ evolves according to the master equation

$$\frac{\mathrm{d}p_x(t)}{\mathrm{d}t} = \sum_{y \in \mathcal{X}} r(y \rightarrow x)p_y(t) - r(x \rightarrow y)p_x(t), \tag{7}$$

where the sum runs over the neighborhood $\mathcal{X}$ of $x$.

The traditional kinetic approach starts by enumerating the whole space (or a carefully chosen subspace) of structures using `RNAsubopt`. Next, this ensemble is divided into local attraction basins separated from one another by energy barriers. This coarsening is usually done with the tool `barriers`. Then, following the Arrhenius formulation, one simulates a coarse grained kinetics between basins. In contrast, the Metropolis scheme used in our kinetic ansatz is based on the stability difference between structures, which may hide energy barriers. Due to this approximation, we referred to our approach as a *kinetic ansatz*.

## Benchmark dataset

To build the dataset for the folding task application, we started from the `ArchiveII` dataset derived from multiple sources [40–56]. We first removed all the structures with pseudoknots, since the tools considered here do not handle them. Next, we evaluated the structures' energies and removed all the unstable structures (i.e. structures with energies $\Delta G_s > 0$). This dataset is composed of 2, 698 sequences with their corresponding known structures. 240 sequences were found multiple times (from 2 to 8 times); 19 of them were mapped to different structures. For the sequences that appeared with different structures, we picked the structure with the lowest energy. In the end, we obtained a dataset of 2, 296 sequences-structures.

## Structure prediction protocols for benchmarks

To evaluate the structure prediction accuracy of the proposed method, we compared it to two structure estimates: the MFE structure and the ML structure. To compute the MFE structure, we used `RNAfold 2.4.13` with the default parameters. We computed the prediction using `MXfold2 0.1.1` with the default parameters for the ML structure. Therefore, only one structure prediction per sequence for those two methods was used for the statistics.

Two parameters are critical for `RAFFT`, the number of positional lags in which stems are searched, and the number of structures stored in the stack. For our computational experiments, we searched for stems in the $n = 100$ best positional lags and stored $N = 50$ structures. The correlation function $\mathrm{cor}(k)$, which allows to choose the positional lags, is computed using the weights $w_{GC} = 3$, $w_{AU} = 2$, and $w_{GU} = 1$.

To assess the performance of `RAFFT`, we analyzed the output in two different ways. First, we considered only the structure with the lowest energy found for each sequence. This procedure allows us to assess `RAFFT` performance in search of low energy structure only. Second, we computed the accuracy of all $N = 50$ structures saved in the last stack for each sequence and displayed only the best structure in terms of accuracy (`RAFFT*`). As mentioned above, the lowest energy structure found may not be the active structure. Therefore, this second procedure allows us to assess whether one of the pathways constituting the ensemble is biologically relevant.

We used two metrics to measure the prediction accuracy: the positive predictive value (PPV) and the sensitivity. The PPV measures the fraction of correct base pairs in the predicted

structure, while the sensitivity measure the fraction of base pairs in the accepted structure that are predicted. These metrics are defined as follows:

$$PPV = \frac{TP}{TP + FP}, \quad \text{Sensitivity} = \frac{TP}{TP + FN}, \tag{8}$$

where TP, FN, and FP stand respectively for the number of correctly predicted base pairs (true positives), the number of base pairs not detected (false negatives), and the number of wrongly predicted base pairs (false positives). To be consistent with previous studies, we computed these metrics using the `scorer` tool provided by Matthews *et al.* [34], which also provides a more flexible estimate where shifts are allowed.

### Structure space visualization

We used a Principal Component Analysis (PCA) to visualize the loop diversity in the datasets considered here. To extract the weights associated with each structure loop from the dataset, we first converted the structures into weighted coarse-grained tree representation [57]. In the tree representation, the nodes are generally labelled as E (exterior loop), I (interior loop), H (hairpin), B (bulge), S (stacks or stem-loop), M (multi-loop) and R (root node). We separately extracted the corresponding weights for each node, and the weights are summed up and then normalized. Excluding the root node, we obtained a table of 6 features and *n* entries. This allows us to compute a 6 × 6 correlation matrix that we diagonalize using the `eigen` routine implemented in the `scipy` package. For visual convenience, the structure compositions were projected onto the first two Principal Components (PC).

## Results

### Application to the folding task

We started by analyzing the prediction performances with respect to sequence lengths: we averaged the performances at fixed sequence length. Fig 5 shows the performance in PPV and sensitivity for the three methods. It shows that the ML method consistently outperformed `RAFFT` and MFE predictions. A *t*-test between the ML and the MFE predictions revealed not only a significant difference (p-value $\approx 10^{-12}$) but also a substantial improvement of 14.5% in PPV. `RAFFT` showed performances similar to the MFE predictions for shorter sequences; however, `RAFFT` is significantly less accurate for sequences of length greater than 300 nucleotides.

The quality of the trajectories predicted by our framework, and therefore, by our kinetic ansatz depends on the quality of the ensemble of structures predicted by the folding component of our method. Consequently, we try here to answer the question: are there relevant structures in the ensemble predicted by our method? To address this question we retained the structure with the best score among the 50 recorded structures per sequence. We obtained an average PPV of 57.9% and an average sensitivity of 63.2% over all the dataset. The gain in terms of PPV/sensitivity is especially pronounced for sequences of length ≤ 200 nucleotides, indicating the presence of biologically more relevant structures in the predicted ensemble than the thermodynamically most stable one (PPV was = 79.4%, and sensitivity = 81.2%). The average scores are shown in Table 1. We also investigated the relation to the number of bases between paired bases (base pair spanning), but we found no striking effect, as already pointed out in one previous study [58].

All methods performed poorly on two groups of sequences: one group of 80 nucleotides long RNAs, and the second group of around 200 nucleotides (two of these sequences are shown in Fig A in S1 Appendix). The PCA analysis of the known structure space, shown in Fig

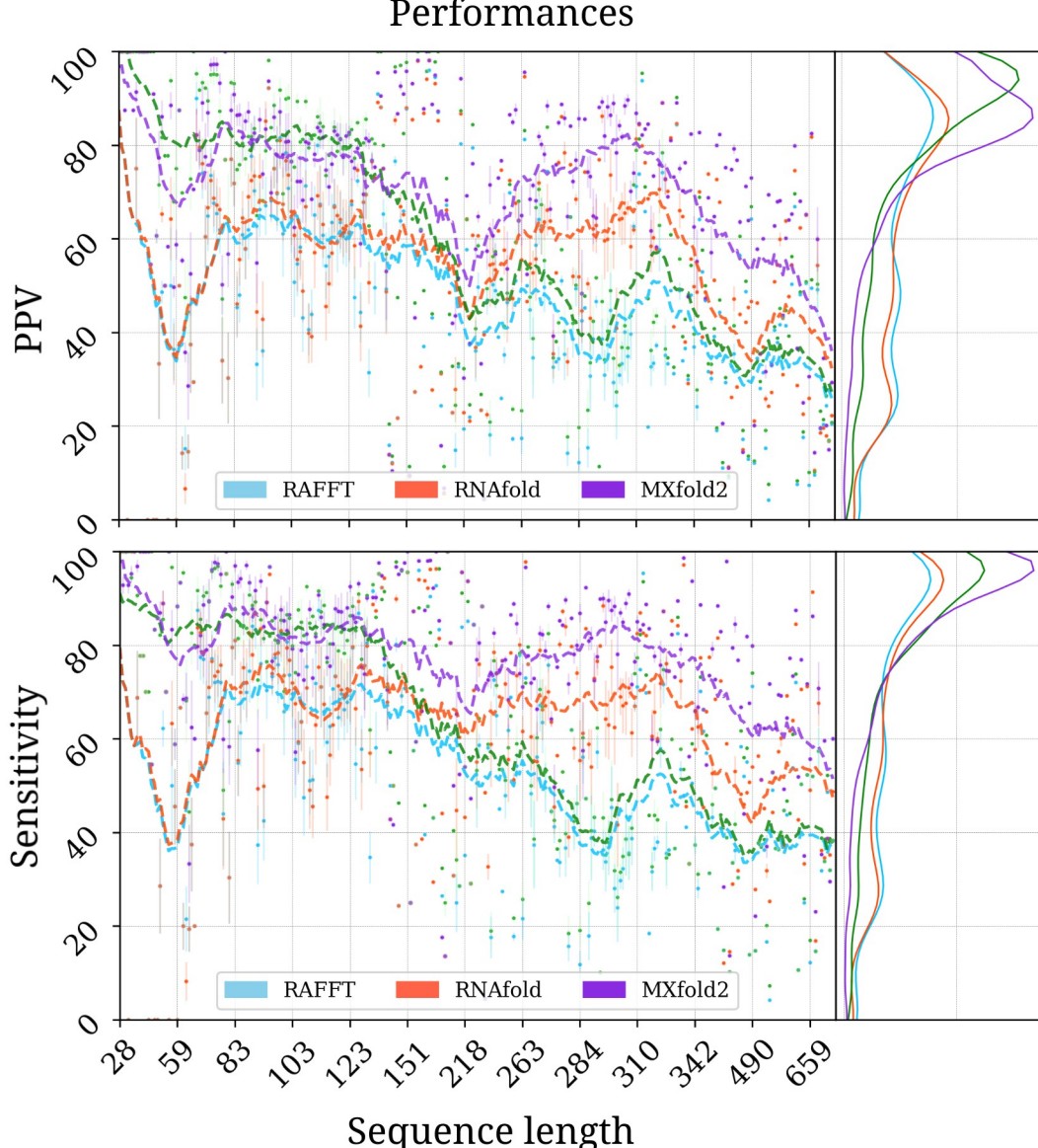

**Fig 5. `RAFFT`'s performance on folding task.** PPV and sensitivity *vs* sequence length. In the left panels, `RAFFT` (in blue) shows the scores when for the structure (out of $N = 50$ predictions) with the lowest free energy, whereas `RAFFT`* (in green) shows the best PPV score in that ensemble. Each dot corresponds to the mean performance for a given sequence length, and vertical lines display their standard deviation. The right panels of both figures show the distribution of PPV and sensitivity sequence-wise.

6, reveals a propensity for interior loops and the presence of large unpaired regions like hairpins or external loops. The structure space produced by the ML predictions seems closer to the native structure space. In contrast, the structure spaces produced by `RAFFT` and `RNAfold` (MFE) are similar and more diverse.

## Test case: The investigation of the CFSE folding dynamics

We applied the RAFFT framework (folding + kinetics) to the CFSE, a natural RNA sequence of 82 nucleotides, where the structure has been determined by sequence analysis and obtained

**Table 1. Average performance displayed in terms of PPV and sensitivity.** The metrics were first averaged at fixed sequence length, limiting the over-representation of shorter sequences. The first two rows show the average performance for all the sequences for each method. The bottom two rows correspond to the performances for the sequences of length ≤ 200 nucleotides. For the ML and MFE only one prediction per sequence and for RAFFT 50 predictions per sequence were used. Here RAFFT (respectively RAFFT*) refers to the case when the lowest free energy (resp. highest PPV) from the ensemble of 50 predictions is selected.

| | RAFFT | ML | MFE | RAFFT* |
|---|---|---|---|---|
| | All sequences | | | |
| PPV | 47.7 | 70.4 | 55.9 | 60.0 |
| Sensitivity | 52.8 | 77.1 | 63.3 | 62.8 |
| | Sequences with lengths ≤ 200 | | | |
| PPV | 57.9 | 76.7 | 59.5 | 79.4 |
| Sensitivity | 63.2 | 82.9 | 65.5 | 81.2 |

from the RFAM database. This structure has a pseudoknot which is not taken into account here.

Fig 7A and 7B respectively show the fast-folding graph constructed using RAFFT, and the MFE and native structures for the CFSE. The fast-folding graph is computed in four steps. At each step, stems are constructed by searching for $n = 100$ positional lags and, a set of $N = 20$ structures (selected according to their free energies) are stored in a stack. The resulting fast-folding graph consists of 68 distinct structures, each of which is labelled by a number. Among the structures in the graph, 6 were found similar to the native structure (16/19 base pairs differences). The structure labelled "29" in the graph leading to the MFE structure "59" is the $9^{th}$ in the second stack. When storing less than 9 structures in the stack at each step, we cannot

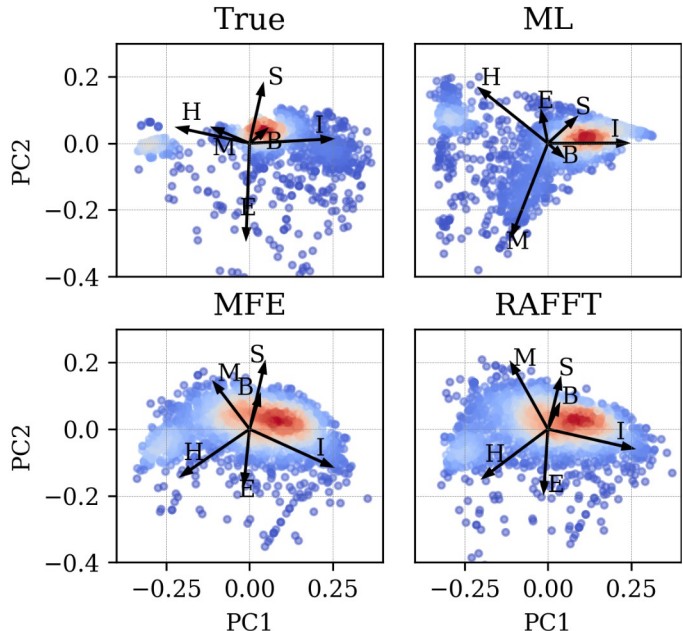

**Fig 6. Structure space analysis.** PCA for the predicted structures using RAFFT, RNAfold, MxFold2 compared to the known structures denoted "True". The arrows represent the direction to secondary structure types (H = hairpin, I = E = exterior loop, I = interior loop, H = hairpin, B = bulge, S = stacks, M = multi-loop and R = root node).

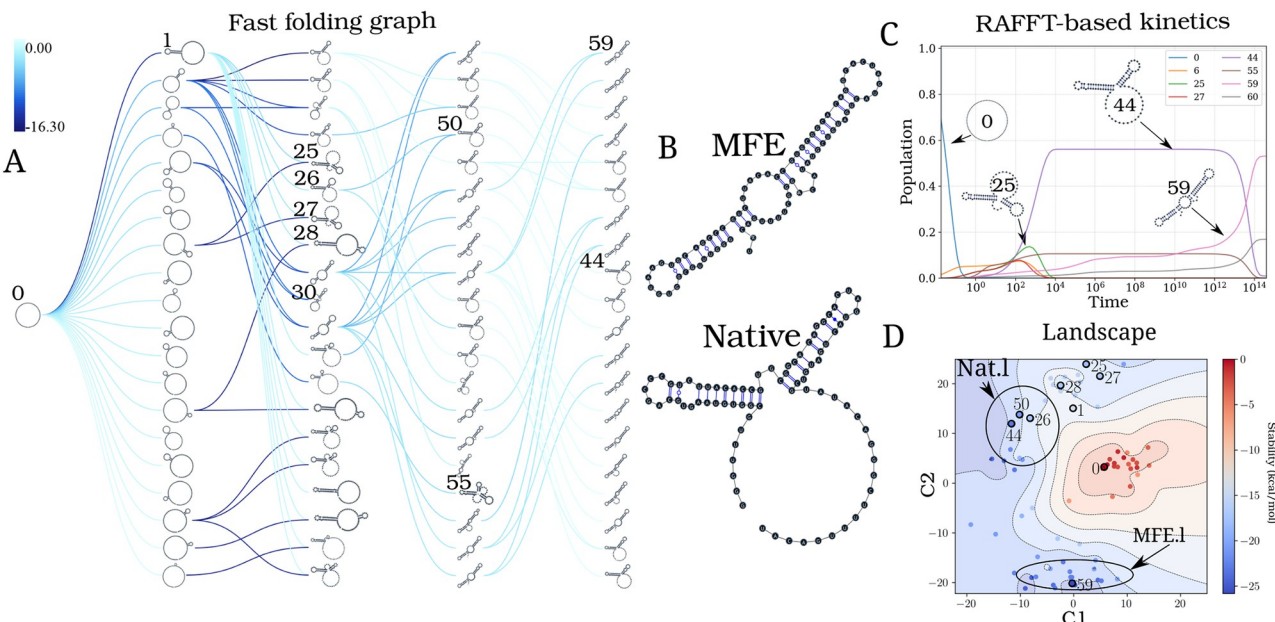

**Fig 7. Application of the folding kinetic ansatz on CFSE.** (A) Fast-folding graph in four steps and *N* = 20 structures stored in a stack at each step. The edges are coloured according to ΔΔ*G*. At each step, the structures are ordered by their free energy from top to bottom. The minimum free energy structure found is at the top left of the graph. A unique ID annotates visited structures in the kinetics. For example, "59" is the ID of the MFE structure. (B) MFE (computed with `RNAfold`) and the native CFSE structure. (C)The change in structure frequencies over time. The simulation starts with the whole population in the open-chain or unfolded structure (ID 0). The native structure (**Nat.l**) is trapped for a long time before the MFE structure (**MFE.l**) takes over the population. (D) Folding landscape derived from the 68 distinct structures predicted using `RAFFT`. The axes are the components optimized by the MDS algorithm, so the base pair distances are mostly preserved. Observed structures are also annotated using the unique ID. MFE-like structures (**MFE.l**) are at the bottom of the figure, while native-like (**Nat.l**) are at the top.

obtain the MFE structure using `RAFFT`; this is a direct consequence of the greediness of the proposed method. To visualize the energy landscape drawn by `RAFFT`, we arranged the structures in the fast-folding graph onto a surface according to their base-pair distances; for this we used the multidimensional scaling algorithm implemented in the `scipy` package. Fig 7D shows the landscape interpolated with all the structures found; this landscape illustrates the bistability of the CFSE, where the native and MFE structures are in distinct regions of the structure space.

From the fast-folding graph produced using `RAFFT`, the transition rates from one structure in the graph to another are computed using the formula given in Eq 6. Starting from a population of unfolded structure and using the computed transition rates, the native of structures is calculated using Eq 7. Fig 7C shows the frequency of each structure; as the frequency of the unfolded structure decreases to 0, the frequency of other structures increases. Gradually, the structure labelled "44", which represents the CFSE native structure, takes over the population and gets trapped for a long time, before the MFE structure (labelled "59") eventually becomes dominant. Even though the fast-folding graph does not allow computing energy landscape properties (saddle, basin, etc.), the kinetics built on it reveals a high barrier separating the two meta-stable structures (MFE and native).

Our kinetic simulation was then compared to `Treekin` [59]. First, we generated $1.5 \times 10^6$ sub-optimal structures up to 15 kcal/mol above the MFE structure using `RNAsubopt` [36]. Since the MFE is $\Delta G_s = -25.8$ kcal/mol, the unfolded structure could not be sampled. Second, the ensemble of structures is coarse-grained into 40 competing basins using the tool `barriers` [59], with the connectivity between basins represented as a barrier tree (see Fig 8A).

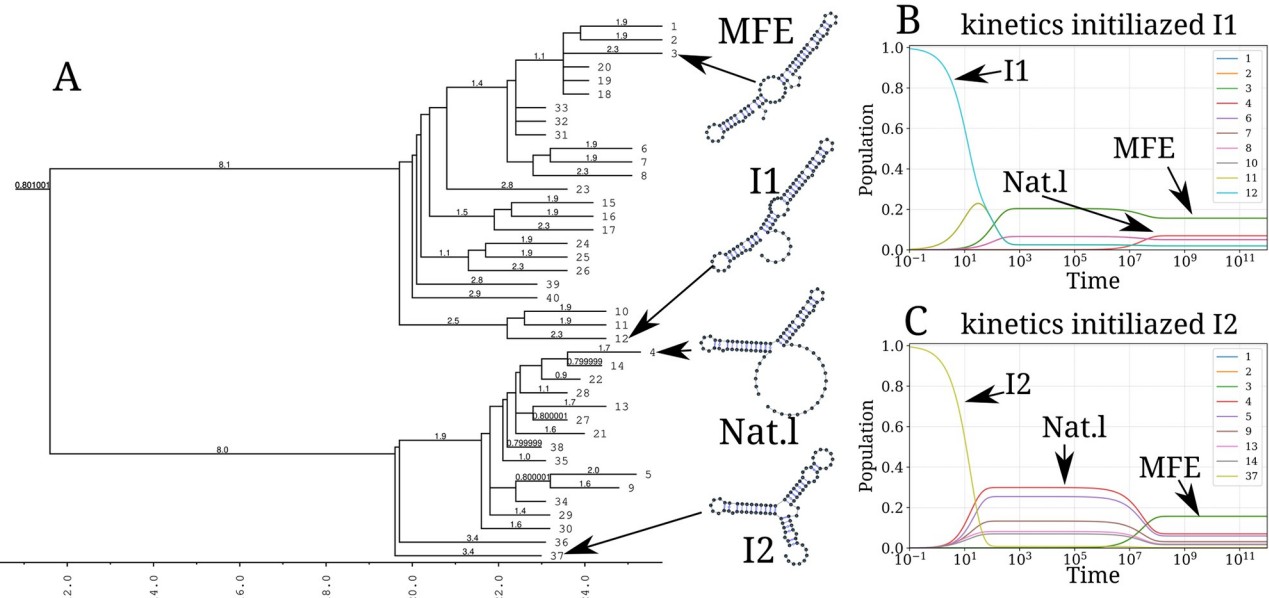

**Fig 8. Folding kinetics of CFSE using `Treekin`.** A) Barrier tree of the CFSE. From a set of $1.5 \times 10^6$ sub-optimal structures, 40 local minima were found, connected through saddle points. The tree shows two alternative structures separated by a high barrier with the global minimum (MFE structure) on the right side. (B) Folding kinetics with initial population $I_1$. Starting from an initial population of $I_1$, as the initial frequency decreases, the others increase, and gradually the MFE structure is the only one populated. (C) Folding kinetics with initial population $I_2$. When starting with a population of $I_2$, the native structure (labelled **Nat.1**) is observable, and gets kinetically trapped for a long time due to the high energy barrier separating it from the MFE structure.

When using `Treekin`, the choice of the initial population is not straightforward. Therefore we resorted to two initial structures $I_1$ and $I_2$ (see Fig 8B and 8C, respectively). In Fig 8B, the trajectories show that only the kinetics initialized in the structure $I_2$ can capture the complete folding dynamics of CFSE, in which the two metastable structures are visible. Thus, in order to produce a folding kinetics in which the native and the MFE structures are visible, the kinetic simulation performed using `Treekin` required a particular initial condition and a barrier tree representation of the energy landscape built from a set of $1.5 \times 10^6$ structures. By contrast, using the fast-folding graph produced by `RAFFT`, which consists only of 68 distinct structures, our kinetic simulation produces complete folding dynamics starting from a population of unfolded structure.

## Discussion

We have proposed a method for RNA folding dynamics predictions called `RAFFT`. Our method was inspired by the experimental observation of parallel fast-folding pathways. `RAFFT` has two components: a folding algorithm and a kinetic ansatz.

First, we showed that our algorithm produces ensembles that contain biologically relevant structures. Two structure estimates were compared to our method: the MFE structure computed using `RNAfold`, and the ML estimate using `MXfold2`. Other thermodynamic-based and ML-based tools were investigated but not shown here because their performances were found to be very similar to the one of `MXfold2` and `RNAfold` (See Fig A in S1 Appendix for the complete benchmark). When we considered the lowest energy structure, the comparison of `RAFFT` to existing tools confirmed the overall validity of our approach. In more detail, comparisons with thermodynamic/ML models yielded the following results. First, the ML predictions performed consistently better than both `RAFFT` and the MFE approach, where the

PPV = 70.4% and sensitivity = 77.1% on average. Second, the ML methods produced loops, such as long hairpins or external loops. We argue that the density of those loops correlates with the ones in the benchmark dataset, which a PCA analysis revealed too. In contrast, the density of loops was lower in the structure spaces produced by RAFFT and MFE, implying some over-fitting in the ML model. Finally, known structures obtained through covariation analysis reflect structures *in vivo* conditions. Therefore, the structures predicted by ML methods may not only result from their sequences alone but also from their molecular environment, e.g. chaperones. We expect the thermodynamic methods to provide a more robust framework for the study of sequence-to-structure relations.

So how does RAFFT predictions contain more relevant structures than the MFE, although these structures are less thermodynamically stable? The interplay of three effects may explain this finding. First, the MFE structure may not be relevant because active structures can be in kinetic traps. Second, RAFFT forms a set of pathways that cover the free energy landscape until they reach local minima, yielding multiple long-lived structures accessible from the unfolded state. Third, the energy function is not perfect, so the MFE structures computed by minimizing it may not be the most stable.

However, identifying these structures in the ensembles produced by RAFFT is not trivial. In contrast to the benchmark data, the native structure is usually unknown, necessitating further analyses of the ensembles output by RAFFT. The empirical results showed that we can use RAFFT fast-folding graph to reproduce state-of-the-art kinetics, at least qualitatively. Our method demonstrated three main benefits. First, the kinetics can be drawn from as few as 68 structures, whereas the barrier tree may require millions. Second, the kinetics ansatz describes the complete folding mechanism starting from the unfolded state. Third, the procedure did not require additional coarse-graining into basins for the length range tested here. (Longer RNAs might require such a coarse-graining step, in which structures connected in the fast-folding graph are merged together).

We believe that the proposed method is a robust heuristic for structure prediction in conjunction with folding dynamics based on our results. The folding landscape depicted by RAFFT was designed to follow the kinetic partitioning mechanism, where multiple folding pathways span the folding landscape. This approach has shown good predictive potential. Furthermore, we derived a kinetic ansatz from the fast-folding graph to model the slow part of the folding dynamics. It approximated the usual kinetics framework qualitatively, requiring drastically fewer structures. Our findings suggest that the RNA folding kinetic partitioning mechanism is indeed following the stem competition at the foundation of RAFFT.

On the one hand, further improvements to RAFFT's algorithm could be investigated:

- the choice of stems is limited to the largest in each positional lag, a greedy choice that may not be optimal. We propose to add stochasticity in the selection of positional lag to keep, such that running multiple times RAFFT, one can overcome some greediness bottlenecks.

- our method constructs parallel pathways leading to a diverse set of accessible structures. Still, we have not given any thermodynamic-based criterion to identify which are more likely to resemble the native structure. We suggest using an ML-optimized score to investigate the restrained ensemble of structures predicted by RAFFT.

- structures connected in the parallel pathways are separated by the formation or unfolding of a single stem. As mentioned above, RAFFT does not account for barriers between structures that stem formation could involve. Therefore, we propose to apply a post-treatment on the folding graph, where the folding path between structures is investigated using the set of valid atomic folding moves (*e.g.* individual base-pair formation).

On the other hand, our method can also find applications in RNA design. The design procedure could start with identifying long-life intermediates and using them as target structures. Moreover, the efficient stem sampling enabled by the FFT can also be straightforwardly applied to the search for RNA-RNA interactions.

## Supporting information

**S1 Appendix. Supporting appendix.** Additional numeric experiments for comparing other folding tools and execution times.
(PDF)

## Acknowledgments

We thank Onofrio Mazzarisi for helpful discussions and Peter F. Stadler for insightful comments.

## Author Contributions

**Conceptualization:** Vaitea Opuu.

**Data curation:** Nono S. C. Merleau, Vincent Messow.

**Formal analysis:** Vaitea Opuu, Nono S. C. Merleau, Vincent Messow.

**Funding acquisition:** Matteo Smerlak.

**Investigation:** Vaitea Opuu, Nono S. C. Merleau, Vincent Messow, Matteo Smerlak.

**Methodology:** Vaitea Opuu.

**Software:** Vaitea Opuu, Nono S. C. Merleau, Vincent Messow.

**Supervision:** Matteo Smerlak.

**Writing – original draft:** Vaitea Opuu.

**Writing – review & editing:** Vaitea Opuu, Nono S. C. Merleau, Vincent Messow, Matteo Smerlak.

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
