## [Decision Letter · Decision Letter 0]

3 Jul 2022

Dear Dr opuu,

Thank you very much for submitting your manuscript "RAFFT: Efficient prediction of RNA folding pathways using the fast Fourier transform" for consideration at PLOS Computational Biology. As with all papers reviewed by the journal, your manuscript was reviewed by members of the editorial board and by several independent reviewers. The reviewers appreciated the attention to an important topic. Based on the reviews, we are likely to accept this manuscript for publication, providing that you modify the manuscript according to the review recommendations.

Sincerely,

Alexander MacKerell

Associate Editor

PLOS Computational Biology

Arne Elofsson

Deputy Editor

PLOS Computational Biology

[LINK]

Reviewer's Responses to Questions

**Comments to the Authors:**

Reviewer #1: In this article, the authors describe a new approach to prediction of RNA secondary structure. Authors have performed a nice literature survey of the previously developed methods and tools before proposing their new approach. The folding algorithm and kinetic ansatz has been developed and described very nicely by the authors. Although ML-based approaches have shown performances much better than RAFFT, the authors have acknowledged the advantages and disadvantages of the method in details. Overall the paper appears to be very well-written and holds the standards of the scientific presentation highly. The paper is very well suited for the journal and will be interesting to the readers. The reviewer only has minor comments.

- Page 1, “... structures; evidence suggests…” Please add the citation.

- Figure 1 - the caption could use more details. Although it is introductory figure, it could also be used to summerize the folding algorithm and kinetic ansatz that will keep readers interested.

- RAFFT - It would be nice if the authors can actually provide a term and abbreviation connection somewhere they first use it.

- Page 3, please merge the sentences “we compute free energy change …” and “Here, free energies were computed”

- Page 3, can others elaborate on the parameterization of the weights.

- Page 3, figure 3 is referred before figure 2. Figures should be numbered in the same order they are cited in the text.

- Page 3, Sentence “Linear fold is the fastest…” seems out of context.

- Figure 6, What do the notations H,M,B etc denote?

Reviewer #2: This manuscript reports a fast method (RAFFT) to generate base-paired stems, ie secondary structure elements, from a nucleotide sequence.

This is fine, but isn't the main issue rather the scoring of the stability of the models?

A "kinetic Ansatz" is proposed as a means of finding optimal structures. The model used does however not account for possibly significant barriers between substates. The issue is recognized, but not resolved in the manuscript. This is a major simplification which could be expected to have a large influence on the results

The basic RAFFT approach performs so-so in terms of accucacy,; when the 50 top-scoring structures are allowed into the game, the chance of finding the "correc" structure increases. But how should the user know which structure(s) to use?

Minor issue:

eq 2 is strange. The right hand side is an illegal matrix multiplication.

**Have the authors made all data and (if applicable) computational code underlying the findings in their manuscript fully available?**

Reviewer #1: Yes

Reviewer #2: Yes

PLOS authors have the option to publish the peer review history of their article (what does this mean?). If published, this will include your full peer review and any attached files.

Reviewer #1: No

Reviewer #2: No

Figure Files:

Data Requirements:

Reproducibility:

References:

---

## [Editor Report · Decision Letter 1]

28 Jul 2022

Dear Dr opuu,

We are pleased to inform you that your manuscript 'RAFFT: Efficient prediction of RNA folding pathways using the fast Fourier transform' has been provisionally accepted for publication in PLOS Computational Biology.

Best regards,

Alexander MacKerell

Associate Editor

PLOS Computational Biology

Arne Elofsson

Deputy Editor

PLOS Computational Biology

---

## [Editor Report · Acceptance letter]

22 Aug 2022

PCOMPBIOL-D-22-00642R1 

RAFFT: Efficient prediction of RNA folding pathways using the fast Fourier transform

Dear Dr opuu,

I am pleased to inform you that your manuscript has been formally accepted for publication in PLOS Computational Biology. Your manuscript is now with our production department and you will be notified of the publication date in due course.

With kind regards,

Zita Barta
